# Wristbands Containing Accelerometers for Objective Arm Swing Analysis in Patients with Parkinson’s Disease

**DOI:** 10.3390/s20154339

**Published:** 2020-08-04

**Authors:** Domiciano Rincón, Jaime Valderrama, Maria Camila González, Beatriz Muñoz, Jorge Orozco, Linda Montilla, Yor Castaño, Andrés Navarro

**Affiliations:** 112t Research Group, Universidad Icesi, Cali 760031, Colombia; domiciano.rincon@correounivalle.edu.co (D.R.); lindagm0319@gmail.com (L.M.); yorjaggy@gmail.com (Y.C.); 2Centro de Investigaciones Clínicas (CIC), Fundación Valle del Lili, Cali 760032, Colombia; jaime.valderrama.chaparro@gmail.com (J.V.); gonzalezv.mariacamila@gmail.com (M.C.G.); 3Department of Clinical Neurophsicology, Fundación Valle del Lili, Cali 760032, Colombia; beatriz.munoz@fvl.org.co (B.M.); jorge.orozco@fvl.org.co (J.O.)

**Keywords:** gait, wearables, mobile health, accelerometer, Parkinson’s disease, age-related chronic diseases/syndromes

## Abstract

In patients with Parkinson’s disease (PD), arm swing changes are common, even in the early stages, and these changes are usually evaluated subjectively by an expert. In this article, hypothesize that arm swing changes can be detected using a low-cost, cloud-based, wearable, sensor system that incorporates triaxial accelerometers. The aim of this work is to develop a low-cost, assistive diagnostic tool for use in quantifying the arm swing kinematics of patients with PD. Ten patients with PD and 11 age-matched, healthy subjects are included in the study. Four feature extraction techniques were applied: (i) Asymmetry estimation based on root mean square (RMS) differences between arm movements; (ii) posterior–anterior phase and cycle regularity through autocorrelation; (iii) tremor energy, established using Fourier transform analysis; and (iv) signal complexity through the fractal dimension by wavelet analysis. The PD group showed significant (p < 0.05) reductions in arm swing RMS values, higher arm swing asymmetry, higher anterior–posterior phase regularities, greater “high energy frequency” signals, and higher complexity in their XZ plane signals. Therefore, the novel, portable system provides a reliable means to support clinical practice in PD assessment.

## 1. Introduction

Parkinson’s disease (PD), characterized by bradykinesia (global slowing of movement), resting tremor, postural instability, and rigidity, is the second most common neurodegenerative disorder [1]. These symptoms of PD usually alter arm and leg movements, leading to a modified gait pattern. Although gait in patients with PD is influenced mainly by changes to the kinematics of the lower limbs, subtle motor changes in arm swing have been described, even during the early stages of the disease [2], which can be relevant to diagnosis and follow-up.

Motor changes are usually evaluated by means of clinical scales, such as the Movement Disorders Society Unified Parkinson Disease Rating Scale (MDS-UPDRS Part III). Although this approach is useful, it is highly subjective [3]. Over recent decades, multiple technological devices have been used to quantify changes to motor control in patients with PD [4], and some of these technologies include body markers [5,6], portable accelerometers, inertial measurement units (IMUs) [7], and depth sensor cameras [8]. However, most of the technologies reported in the literature have been used only in a research context.

Body markers are currently one of the most comprehensive tools available for assessing gait in PD patients because, using infra-red cameras, they can capture the movement of every joint using physical references attached to the body. Therefore, the complete kinematics of gait can be evaluated by comparing patients and controls. For example, Carpinella et al. [5] used cameras to determine movement in PD patients and control subjects wearing retroreflective markers. They showed that patients in the early stages of the disease (Hoehn and Yahr stages I and II) show reduced walking speeds, cadence, and turning capabilities. Furthermore, focusing on the arms, Roggendorf et al. [6] measured limb kinematics in patients with early-stage PD using ultrasound-based body markers. The PD group showed a significant reduction in the swing amplitude of the arm on the most affected side of the body, and this difference in measurements between arms concurs with the asymmetry index, which is higher in PD patients compared with controls. However, assessments using body markers are available only in gait laboratories, where experiments are elaborate and expensive, and not in ambulatory care centers. For this reason, we saw a requirement to endorse the use of low-cost sensors, capable of confirming cinematic differences between PD patients and controls.

Accelerometers are low-cost devices, which can provide relevant information to physicians in a clinical context. Although IMUs are more precise, they cost more, which can be problematic to the health systems of middle-income countries. Accelerometers are widely used, from simple step counters in smartphones to more complex signal processing tools applied in gait and arm swing analyses [9,10,11]. For instance, in the study by Yang et al. [12], a single tri-axial accelerometer was used in real-time gait cycle parameter recognition. For this purpose, single sensors were placed at the waists of PD patients and healthy individuals. Differences were found when comparing gait variables, such as stride length, step time, step length, and step time asymmetry. The group concluded that a single accelerometer-based sensor can be useful for assessing mobility and ambulatory follow-up.

Nevertheless, there is a drawback to the use of accelerometers. These devices not only quantify linear acceleration, but also determine translational and gravitational acceleration, and these three signals cannot be separated. Hence, the double-sensor algorithm was proposed by Liu et al. [13], in which two accelerometers were located on a limb at a known distance apart, to eliminate the translational, and gravitational effects by obtaining the difference in the signals acquired by both accelerometers. This method was then used by Huang et al. [14] on upper extremities to measure gait parameters, such as asymmetry in the arm swing (ASA), where they found significant differences between PD patients and controls.

As a result of the ability of accelerometers to react to incoming signals, tremors have also been studied using these devices. In a study conducted by Joundi et al. [7], a smartphone’s built-in accelerometer was used to evaluate the types of tremor, confirming that the fundamental frequencies of tremors can be compared with those measured using EMG analysis. More recent works use accelerometers based in smartwatches, as well as accelerometers in wrists, feet and legs [15,16,17,18,19,20]. From those references, the most similar to our work is Reference [15], but in this paper, the authors use Machine Learning techniques to analyze the data, and they do not make a clinical analysis. Other interesting references to mention are References [21,22,23], which are systematic revisions of gait analysis using different wearable devices, like accelerometers but mainly inertial units. In Reference [22], the –authors analyze publications from January 1st, 2005 until December 31st, 2017. In Reference [21], the authors analyze papers proposing the use of inertial sensors to predict falling risk. In Reference [23], the authors focus on the use of raw accelerometers data with Machine Learning algorithms, to determine if those techniques are suitable for human activity identification.

The appraisal of motor impairment is extremely useful when evaluating disease progression in, and pharmacologic responses of patients with PD, and limitations to this occur because of subjectivity when applying clinical assessment scales. The aim of this work is to demonstrate the use of accelerometers in a clinical context to support neurologists in the diagnosis and follow-up of patients with PD. Moreover, we aim to develop the use of portable accelerometers as a good option for detecting and quantifying motor alterations not only because of their low-cost but also because of their size, portability, and precision. Hence, this work focused on establishing the feasibility of using low-cost accelerometers in a clinical context to assist physicians in the diagnosis and motor examination of patients with PD. To this end, we implemented and compared four different algorithms based on existing state-of-the-art models and modified these according to our setup to determine the clinical relevance of each. In the following sections, we will show that, although all of the algorithms are technically feasible and show some interesting results, there are aspects, such as clinical meaning and technical complexity, that lead us to choose only a subset of them. The main contribution of this paper is to validate the use of accelerometers located at the wrist in the objective measurement of ASA and other arm swing variables in a clinical context to diagnose and assess patients with early-stage PD.

In order to demonstrate the feasibility and capabilities of the wrist-based accelerometers, we show that the system and the algorithms developed can differentiate healthy subjects from patients with PD, based on their arm swing kinematics, using double, or single accelerometers on each wrist. The remainder of this paper is organized as follows. In Section 2, we describe the materials and methods used, including the selection of patients, the accelerometers and signal recording system, and the algorithms implemented. In Section 3, we describe the results. In Section 4, we discuss the clinical and technical perspectives of the results; and in Section 5, we provide conclusions and possibilities for future work.

## 2. Materials and Methods

### 2.1. Patient Selection

Patients were recruited during neurological consultations at the Fundación Valle del Lili. All patients were able to walk unassisted. All participants in the PD group were treated with dopaminergic agonists and evaluated while in the “ON” state. The “ON” state refers to times when symptoms are better controlled, and the “OFF” state refers to the reappearance or increase of motor symptoms [24]. Patients were age-matched healthy control subjects. This research was approved by the Institutional Review Board of the Fundación Valle del Lili (project number 1146, issued on May 31, 2017), in accordance with the Declaration of Helsinki.

### 2.2. Measurement System

A set of two wristbands containing portable accelerometers was developed for the experiment. Each band consisted of two accelerometers (ADXL335, three-axis sensors with a resolution of ±3 × g) assembled on two Arduino Simblee Bluetooth low energy programmable cards, each one connected to a lithium battery with a capacity of 400 mAh (up to 12 h of use). The programmable card was chosen based on its size, Bluetooth 4.0 connectivity capacity, low power consumption, and resolution (10-bit analog-to-digital converter ports).

The card was programmed in such a way that the data collection process could be activated/deactivated remotely using a mobile application. The sampling frequency for each measurement was 50 Hz. This configuration allowed a wireless network interface controller to pass the information obtained from the accelerometers and send it, via Bluetooth, to a smartphone, from where the information was uploaded to a database on a cloud-based system (Figure 1).

### 2.3. Experimental Setup

All subjects attended a single gait analysis session to determine the movement of each arm using the accelerometer wristbands. Each subject was instructed to walk six times at normal pace along an aisle 10 m long × 1.5 m wide. The aisle length was determined by bearing in mind the minimum distance over which arm swing movement reaches a steady-state (constant speed). A sampling frequency of 50 Hz was set as being adequate for the estimation of arm dynamics [24]. 

### 2.4. Feature Extraction Techniques

Four feature extraction techniques were applied. These were adapted from previous studies in which accelerometers were used. Technique A–called “Asymmetry Calculation”–was based on the results of a study by Huang et al. [12], although in the present study, a shorter capture time suitable for an ambulatory context was used. Technique B–called “Gait Regularity”–was based on a study by Yang et al. [13], in which, unlike in the present study, the sensor was located in the lumbar area. Technique C–called “Energy Distribution”–was based on a fast Fourier transform (FFT). Finally, Technique D was based on the results of a study by Sekine et al. [14], in which they characterized accelerometer signals obtained from the lumbar area through a wavelet transform. In this study, the sensors were located on the wrists. The four techniques are described in detail below.

#### 2.4.1. Root Mean Square (RMS) and RMS Asymmetry Calculations

This technique was based on previous works by Huang et al. [14] used RMS to calculate ASA and differentiate PD patients from controls using a wristband with two accelerometers on each wrist, separated by a fixed distance (Figure 1). In order to obtain the RMS and RMS asymmetry of the signals from each arm, we calculated the angular acceleration using Equation (1).
(1)$$\alpha_{axis}=\frac{A_{axis,low}-A_{axis,high}^{}}{L}$$
where L = 0.1 indicates the distance between the accelerometers attached to each arm in meters and *A_axis,low_* and *A_axis,high_* correspond to the values obtained from the closest (low) and farthest (high) accelerometers from the wrist (Figure 1).

Then, angular acceleration magnitude was estimated, the vertical offset was removed, and the RMS was extracted from the resultant signal. This technique involved the calculation of the difference in the RMS from the acceleration magnitudes of each arm. A higher *ASA* value indicates greater asymmetry. We determined the RMS asymmetry (ASA) using Equation (2).
(2)$$ASA=\frac{45{}^{\circ}-\arctan\left({\mathrm{RMS}}_{\min}/{\mathrm{RMS}}_{\max}\right)}{45{}^{\circ}}*100,$$
where RMS_min_ and RMS_max_ are the minimum and maximum effective values obtained during one gait trial. Figure 2 shows an example of an accelerometer signal with an RMS value.

#### 2.4.2. Arm Swing Cycle Regularity and Arm Swing Posterior–Anterior Phase Regularity

This method was used by Yang et al. [25] and Moe-Nielsen et al. [26] to describe stride and step regularity based on accelerometers attached to the base of the spine. We moved the accelerometers to the wrist and modified the signal-processing algorithm in order to analyze arm swing with good results.

The arm swing cycle and arm swing posterior–anterior phase regularity were calculated using a correlation analysis. The autocorrelation allowed us to find repetitive patterns and measure the regularity of arm signals. Considering the fact that arm signals are quasiperiodic, the patterns allowed us to find signal periodicity in order to obtain the dominant frequency of the arm swing. We applied a normalized unbiased autocorrelation function to the *Y*-axis signal from the accelerometer placed closest to the wrist on each arm. Samples of registered signals from a patient and control are shown in Figure 3.

The locations of peaks A and B on the time axis of the resulting signal match the first and second dominant periods, respectively. The amplitude of the first peak (A) represents the autocorrelation between the posterior and anterior phases of the arm swing (Figure 4) in each arm swing cycle, and peak B represents the correlation between the total arm displacements between different cycles for the same walking trial. In a normal walk, there is a difference between the amplitudes of the front and rear swing, and patients with PD tend to exhibit similar amplitudes in both front and rear swing.

#### 2.4.3. Arm Swing Energy Distribution

This method used the FFT of each measurement axis to obtain the amplitude of the signal’s FFT and its frequency distribution. With this method, tremors can only be detected in patients with it. It is well known that PD tremor occurs in the range of 5–10 Hz. Considering that a high-frequency upper extremity tremor is one of the main clinical features of PD, an FFT was used to calculate the energy of the signal from the closest accelerometer of each arm. To do so, we calculated the FFT for each accelerometer axis by estimating the magnitude of the resultant signal from the integration of the vectors. The energy was assigned as low, mid, or high according to the frequency range distribution: Low Energy—the sum of the FFT components was between 0 and 2.5 Hz; Mid Energy—the sum of the FFT components was between 2.5 and 5 Hz; and High Energy—the sum of the FFT components was between 5 and 10 Hz. Total energy was defined as the sum of the FFT components between 0 and 10 Hz. Figure 5 shows the differences in energy range and frequency distribution between a healthy subject and a PD patient with a tremor in the left arm—peaks in the 5–10 Hz band.

#### 2.4.4. Fractal Dimension of Arm Swing Signals

This approach was based on the work of Sekine et al. [27] that describes the relationship between the variance and the density of the power spectrum of a signal using the wavelet transform. In the original work, the authors placed the accelerometer in the subject’s back to analyze gait activities, including walking and stair climbing. We placed the accelerometers on the wrists to analyze arm swing and used signal complexity to identify PD characteristics. We calculated the fractal dimension of the arm swing signals using a wavelet transform. In order to do this, we first applied the empirical equation Equation (3), used by Sekine et al. [27], which defines the power spectrum density of signal in terms of variance and frequency:(3)$$S\left(w\right)∼\frac{\sigma^{2}}{{\left|w\right|}^{\beta }}$$
where S(w) is the power spectrum density, w is the frequency, ß is the spectral component, and σ^2^ is the variance of the original signal. Then, using the wavelet transform, the original signal is decomposed in d_j_ signals. Because each of the d_j_ detail signals extracted along the decomposition levels has a frequency band f_s_/2^j+1^ < w < f_s_/2^j^ (where f_s_ is signal sample rate), Equation (3) can be rewritten according to Reference [27] as Equation (4), which relates the spectral component (ß) with the total variance of the signal (σ^2^) and each variance of detail signal (d_j_), after being decomposed via a wavelet transform:(4)$$Variance\left(d_{j}\right)∼\frac{\sigma^{2}}{{\left(2^{j}\right)}^{\beta }}$$
where j is the level of wavelet decomposition; d_j_ is each detailed signal produced by wavelet transform at level j. Applying log2 scale on Equation (4), the expression becomes a straight line where ß is the slope, being $\sigma^{2}$ not a function of level j. As shown in Figure 6, effectively the logarithmic scale scatter plot of detailed signal variance at level j along level j of decomposition, shows a linear trend.

In our case, wavelet decomposition was performed using Daubechies 4 wavelets (db4) with five levels of detail on each of the three signal axes (X–Y–Z) for the accelerometer located closest to the wrist on each arm. Figure 6 shows that there is a difference in the slopes for each of the patient’s arms, whereas the slopes for the control were nearly identical. This result demonstrated that it is possible to identify a patient with PD because of the difference in the signal variance of each arm, which is equivalent to the asymmetry shown by the RMS value mentioned previously.

Finally, using the value of the spectral component extracted through linear regression, a fractal dimension value was calculated using Equation (5). Hurst exponent (H) is calculated using the spectral component (β) and relates directly to fractal dimension D.
(5)$$D=2-H=2-\frac{\beta -1}{2}$$

The fractal dimension (D) and the spectral component (β) have an inverse linear relationship. A higher spectral component (Figure 6) means a smaller value of the fractal dimension. D value has a range 1 < D < 2, that means that the spectral component (β) has a range 1<β<3, following Equation (5). Finally, as the D value approaches 1 (β value approaches 3) the signal shape is smooth, and as the D value approaches 2 (β value approaches 1), the signal shape is complex. 

In the following sections, we use (D) as the relevant indicator and explain further details about its clinical meaning.

### 2.5. Clinical Evaluation

A movement disorder specialist confirmed the diagnosis of PD according to the UK Parkinson’s Disease Society Brain Bank Diagnostic Criteria and performed MDS-UPDRS Part III. This scale evaluates the magnitude of motor symptoms in PD patients; the higher the score, the higher the motor impairment. A clinical examination was undertaken to establish that there was no other clinical condition present in each patient that could affect gait and arm swing performance. A MoCA test was applied as a cognitive screen to all subjects. To evaluate the motor differences between limbs, asymmetry was defined as the difference between the summed UPDRS scores of the left and right arms (i.e., UPDRS scores 3.3–3.6 and 3.15–3.17). The side (*hemi-body*) with a greater *assessment* score was defined as the most affected.

### 2.6. Statistical Analysis

As the most affected side of a patient is not the same in every case, data were renamed as the less affected side (LAS) and most affected side (MAS) according to the results of the MDS-UPDRS. Continuous arm swing variables were compared using a Mann–Whitney U or “Z” test based on their normality distribution according to Shapiro–Wilk test. A statistically significant difference was indicated by a p-value ≤ 0.05. Statistical analyses were performed using MATLAB^®^ 2018a software.

## 3. Results

Ten patients with PD and 11 age-matched healthy participants (controls) were recruited for the study. The median age of the group was 61 years [inter-quartile range (IQR): 51–74 years; PD: 63.5 years, IQR: 53–76 years; healthy subjects: 60 years, IQR: 51–66 years; p = 0.326). The median duration of the disease from onset was five years (IQR: 4–5 years). The Hoehn and Yahr stage classifications were I (10%), 1.5 (20%), and II (70%). The mean MDS-UPDRS score was 25.8 (±10.27) according to the UPDRS summed scores. Five patients had left-sided PD (50%), four had right-sided PD (40%), and one patient had symmetrical PD. Nine of 10 patients had MoCA test results (total score: 25, IQR: 19–26). Table 1 summarizes the results of the arm swing variables. In Table 1, we use the acronyms LAS and MAS referring to Less Affected Side and Most Affected Side, respectively, to avoid confusions regarding the patients with left or right side affected. Moreover, the terms Dx, Dy, and Dz refer to the fractal dimension in x, y, and z axis, respectively. Axis x and z are statistically relevant in the fractal dimension for both sides (Most Affected and Less Affected). 

## 4. Discussion

The purpose of this research is to develop and validate a clinical support tool using wristbands with accelerometers to quantify the arm swing kinematics of patients with PD. According to our results, PD patients showed significantly lower RMS (LAS, *p* < 0.05; MAS, *p* < 0.05) and higher ASA RMS (*p* < 0.05) values compared to controls. Previous studies have established that PD patients move their arms more slowly and asymmetrically than healthy subjects [28]. Given that RMS is an indicator of signal power and is directly correlated with the magnitude of the angular acceleration of the arm, we believe that RMS may be associated with the distance traveled by the wrist in an arm swing cycle. The lower the RMS, the lower the magnitude of the arm movement. Attempts to integrate accelerometer signals to obtain values for arm swing speed and angle were also made, but the outcomes were inaccurate and did not offer sufficient information to obtain reliable data.

The relationship between the anterior and posterior phases of an arm swing was explored using the signal’s regularity components. It is known that under normal conditions, the anterior displacement during an arm swing is larger than the posterior displacement. According to our results, PD patients exhibited lower irregularities in their anterior–posterior signal relationship (LAS, *p* = 0.042; MAS, *p* = 0.046), suggesting that there was a shortening in the length of the anterior arm swing displacement that made the anterior and posterior phases more alike. This “shortening” of the anterior signal might be explained by the clinical manifestation of stiffness in the upper limbs [29]. No significant differences were found for arm swing cycle regularity. Previous reports have suggested that PD patients have higher variability in their step length [30,31], but our results indicated that such findings might differ according to arm swing cycle variables.

There were no statistically significant differences when comparing total signal energy in our groups. PD patients displayed higher signals above 5 Hz (LAS, *p* < 0.05; MAS, *p* < 0.05). This finding supported the idea that PD patients have faster dynamics (tremor, “high” energy band) that are superimposed on the normal frequency signals (“mid” and “low” energy bands) of arm swing.

When the signals were discomposed in their three axes, and the fractal dimension components were analyzed, significant differences between patients and controls were found in the X- and Z-axes. Although the primary anterior–posterior arm swing movement was executed in the XY plane, the “*coin*-*counting*” tremor was observed mainly as a rotational movement in the XZ plane (Figure 7). As expected, the tremor contributed to an increase in the complexity of the signal, as measured by the fractal dimension. This finding explained why the patient’s X- and Z-axes displayed higher complexity levels in both arms. It is important to note that the scale showed that the difference in the complexity was on the order of decimals.

The devices used in this study were capable of detecting the bradykinetic gait, referred to by Carpinella et al. [5], where PD patients effectively displayed a decrease in arm swing, unlike the controls, as evidenced by the RMS value. Additionally, we were able to strengthen the evidence reported by Roggendorf et al. [6], who established that the bilateral decrease in retroversion in PD patients is related to the regularity of the posterior–anterior phase. In terms of asymmetry, the accelerometers recorded an average 2:1 ratio in the asymmetry index, unlike the 3:1 ratio found by Roggendorf et al., not only because of the sensor components, but also because our calculated ASA was obtained from the values for acceleration and not from the position of the limbs.

Furthermore, the methods and techniques used to calculate the ASA value in this study were the same as those used by Huang et al. [14]. However, our findings demonstrated that the ASA measurement was possible during a 10 s gait assessment using space captures of at least 10 m while reaching the steady-state of the gait, unlike Huang et al., who used an 8 min gait measurement.

### Advantages, Limitations, and Future Work

In terms of innovation, our wristband prototype has taken advantage of the characteristics of portability and connectivity available in personal smartphones by establishing a communication architecture that uses them as intermediates between the sensors and the cloud, from where the data are visible to any analysis software. It should be pointed out that the proposed architecture is scalable, with a design-oriented to web-based services, implying that the devices can be used to acquire samples in a normal clinical environment and the physician can then access the information at any time.

Most studies use two accelerometers to characterize limb movement, which implies that the use of commercial devices is limited, and implementation of signal processing techniques is more complex. The results obtained using the ASA are relevant from the clinical point of view and cheaper than using IMUs. Using a single accelerometer in each wrist, we obtained variables, including arm swing regularity, energy, and fractal dimension, that allowed us to differentiate patients with PD from controls. In the near future, simplification of the signal capture method will enable the use of commercial devices integrated into our cloud-based architecture.

Although significant differences have been identified, a larger sample is needed to confirm our findings. In future studies, features (such as demographic and anthropometric characteristics) must be included in the description of the study population. One has to be cautious and circumspect when evaluating results that limit the potential impact of demographic and anthropometric characteristics. Nevertheless, these details are often missing from the literature because of the methodological difficulties that they may represent.

The estimation of variables, such as angular speed and angle magnitude, was limited, and these restrictions are inherent to the exclusive use of accelerometers without any inertial reference (i.e., IMUs). In future work, IMUs (a gyroscope attached to a magnetometer and an accelerometer) could be integrated into the system to facilitate the estimation of some spatiotemporal variables, thus improving the accuracy of the measurements through the synchronization and correction of readings obtained from the individual signals of the three sensors [32]. Although the IMUs can be useful in the clinical environment, the cost may restrict their regular use, hence making the use of bracelets containing accelerometers a more promising option.

## 5. Conclusions

Patients with PD displayed significantly lower RMS values, higher asymmetry, lower posterior–anterior phase regularity, greater “high” energy band signals, and more complex signals than the control subjects. These variables can easily be obtained using multiple processing techniques applied to signals from triaxial accelerometers obtained in a relative short capture area (10 m) during a medical consultation. Although the results show the promising use of such devices in clinical and daily life environments, the regulatory process can be complex.

In this work, we developed an accelerometer-based system by modifying algorithms and processing techniques used to analyze arm swing. Clinical information related to PD was extracted to obtain quantitative information to support the clinical assessment of patients, complementing traditional subjective techniques. This novel accelerometer-based system can easily be used in the clinician’s office and does not require additional time or complex settings to evaluate patients.

## Figures and Tables

**Figure 1 sensors-20-04339-f001:**
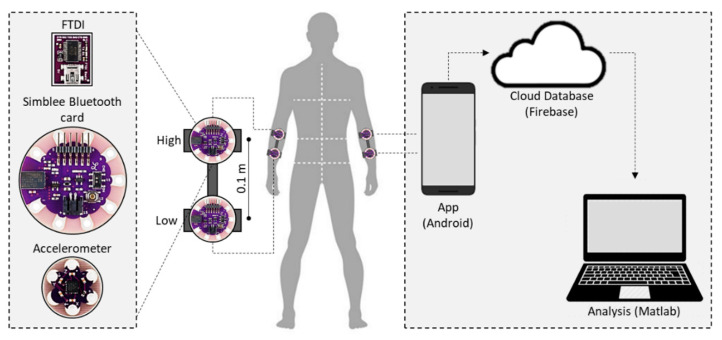
Smart wristbands used to measure the linear and angular acceleration of the subject’s arms in three axes. Data are uploaded through a network consisting of a smartphone that receives the data and then uploads it to a database on the cloud for subsequent analysis using a MATLAB^®^ script.

**Figure 2 sensors-20-04339-f002:**
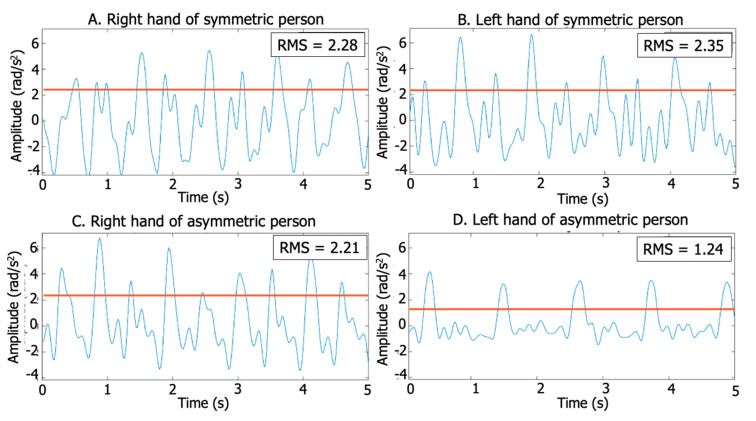
Acceleration signal and RMS for an PD patient and a control. (**A**) Control right arm swing. (**B**) Control left arm swing. (**C**) PD patient right arms swing. (**D**) PD patient left arm swing. Note that the patient exhibits very different RMS, due to waveform and amplitude between limbs.

**Figure 3 sensors-20-04339-f003:**
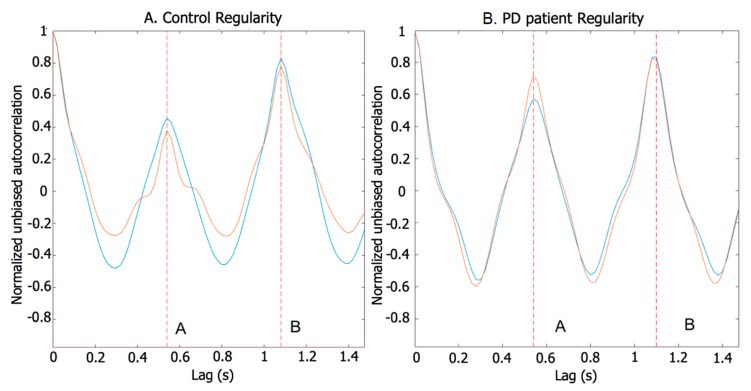
Regularity based on autocorrelation of accelerometer signals. (**A**) Control regularity. (**B**) Patient Regularity. (Red) left arm. (Blue) Right arm. The first peak (A) reveals the regularity of the step, whereas the second peak (B) reveals the regularity of the stride. Patient presents a lower A peak than control one. The B peaks in both cases are similar.

**Figure 4 sensors-20-04339-f004:**
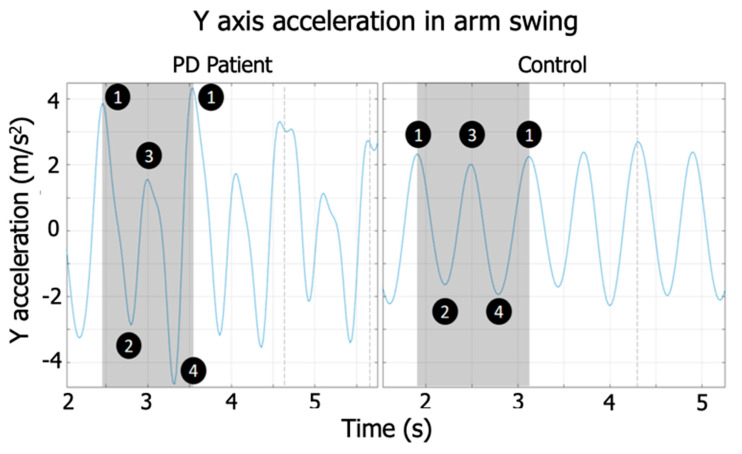
1–2–3–4–1 represents all phases of a complete gait cycle. 1–2–3 are representations of gait states in which the feet and arms have completed a half gait cycle. The peaks in Y are presented in the changes of the direction of arm movement (1, 3, and 1), and their valleys appear when the arm crosses the frontal plane (2 and 4).

**Figure 5 sensors-20-04339-f005:**
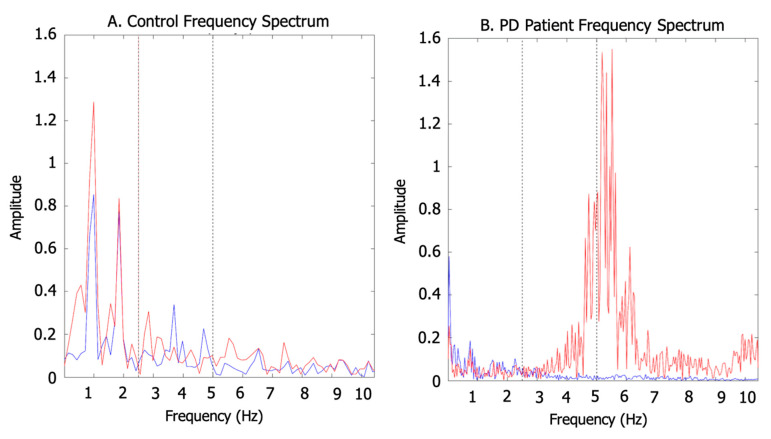
Examples of the spectral frequency for PD patients and controls. (**A**) Control frequency spectrum, step and strike peaks are in 1Hz and 2Hz, respectively. (**B**) Patient frequency spectrum, step and strike peaks are lower than tremor peak around 5H. (Red indicates the left arm, and blue indicates the right arm).

**Figure 6 sensors-20-04339-f006:**
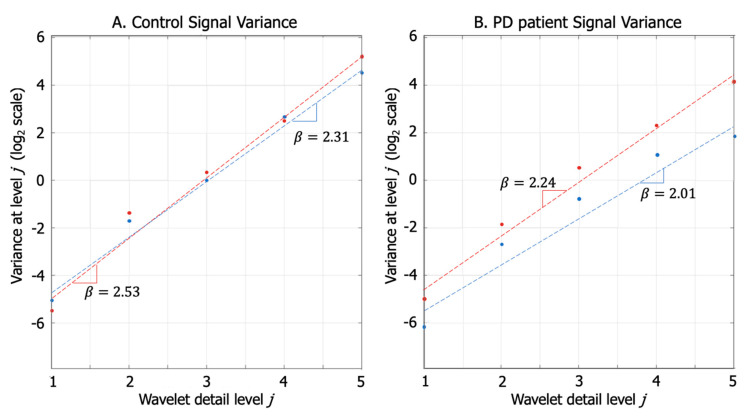
Variance at the wavelet level of detail j (in log_2_ scale) for an PD patient and a control. (Red indicates the left arm, and blue indicates the right arm). (**A**) Control signal variance. Both arms variance through wavelet decomposition present linear trend, which causes a slightly different slope in the line (β). (**B**) PD patient signal variance. The less affected arm has a normal slope (β), but the affected arm presents a lower slope (β = 2.01).

**Figure 7 sensors-20-04339-f007:**
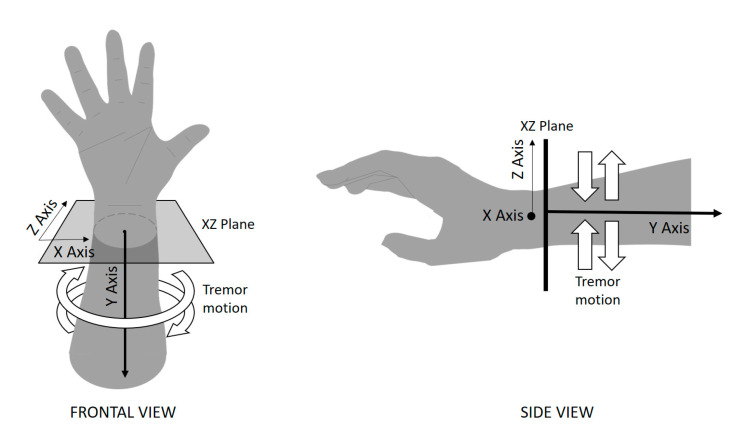
In patients with Parkinson’s disease, a tremor is present in the XZ plane, and the *Y*-axis acts as the axis of rotation for this movement.

**Table 1 sensors-20-04339-t001:** Results of feature extraction techniques for arm swing variables.

Variable	Control Median (IQR)	Patient Median (IQR)	*p*-Value U-Test	Effect Size d Cohen **
RMS LAS ^1^	5.249 (1.052)	4.289 (2.163)	**<0.05 ***	**0.375**
RMS MAS ^1^	4.641 (1.511)	3.493 (2.012)	**<0.05 ***	**0.934**
ASA of RMS ^1^	8.877 (7.849)	16.385 (14.506)	**<0.05 ***	**0.927**
Posterior–anterior phase regularity LAS ^2^	0.512 (0.288)	0.562 (0.148)	**<0.05 ***	**0.378**
Posterior–anterior phase regularity MAS ^2^	0.544 (0.241)	0.629 (0.258)	**<0.05 ***	**0.324**
Arm swing cycle regularity LAS ^2^	0.696 (0.113)	0.705 (0.137)	0.142	0.250
Arm swing cycle regularity MAS ^2^	0.731 (0.119)	0.713 (0.115)	0.359	0.041
Low Energy LAS ^3^	5.393 (2.214)	4.890 (1.882)	0.156	0.351
Mid Energy LAS ^3^	2.907 (1.628)	2.758 (0.769)	0.213	0.093
High Energy LAS ^3^	5.702 (2.242)	6.502 (2.045)	**<0.05 ***	**0.562**
Total Energy LAS ^3^	14.543 (6.108)	14.794 (3.658)	0.276	0.057
Low Energy MAS ^3^	5.321 (1.878)	4.664 (1.393)	0.073	0.459
Mid Energy MAS ^3^	2.974 (1.655)	2.595 (0.473)	0.127	0.224
High Energy MAS ^3^	5.786 (2.240)	7.090 (2.528)	**<0.05 ***	**0.814**
Total Energy MAS ^3^	14.408 (5.825)	14.600 (3.580)	0.210	0.102
Dx LAS ^4^	1.446 (0.230)	1.564 (0.288)	**<0.05 ***	**0.521**
Dx MAS ^4^	1.432 (0.243)	1.607 (0.297)	**<0.05 ***	**1.196**
Dy LAS ^4^	1.529 (0.171)	1.499 (0.329)	0.271	0.042
Dy MAS ^4^	1.506 (0.215)	1.523 (0.302)	0.094	0.163
Dz LAS ^4^	1.291 (0.248)	1.402 (0.210)	**<0.05 ***	**0.516**
Dz MAS ^4^	1.281 (0.179)	1.459 (0.266)	**<0.05 ***	**1.153**

^1^ Variable extracted by the root mean square (RMS) and RMS asymmetry calculation. ^2^ Variable extracted by regularity estimation. ^3^ Variable extracted by arm swing energy distribution. ^4^ Variable extracted by wavelet decomposition applied to x, y and z axis of wrist accelerometer signal. * The variable has a normal distribution. ** The variable presents significant differences between patients and controls. ASA = asymmetry in the arm swing, LAS = least affected side, MAS = most affected side.

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
