# Peer review of "Wristbands Containing Accelerometers for Objective Arm Swing Analysis in Patients with Parkinson’s Disease"

_sensors, 2020, doi:10.3390/s20154339_

Round 1
Reviewer 1 Report
Dear Authors,
I have some comments on your article:
General remarks:
- I have a question, did the authors consider the mathematical description of the movement of a healthy and sick person?
- Description of experimental research and equipment used, measurement techniques should be more accurate.
- Currently, MATLAB was used for the analysis. Are you planning to use other analysis tools?
Detailed comments:
- Sections Measurement System and Experimental Setup should more accurately describe the measurement method and the components of the proposed system.
- In my opinion, it should be in the caption in Figure 2. Describe exactly what is on each of the four component drawings. Enter points a) ...., b)…,
- Similarly, for Figures 3,5, and 6 it is necessary to explain what is on the individual components of the figures.
- Please check the literature, it is formatting, and whether I have newer literature items.
Author Response
General remarks:
- I have a question, did the authors consider the mathematical description of the movement of a healthy and sick person?
Thank you for such interesting question. In this paper we are not considering the mathematical modelling. We analyzed and compared signals between patients and controls using accelerometers on the arms using feature extraction techniques. However, our research group has mathematicians and physicists who are currently mathematically modeling the movement of healthy people and PD patients. We have been working in such modelling using different approaches, like the modelling of a robotic bipedal system, the modelling based on biomechanical analysis used for athletes, until the use of analytics and machine learning techniques to describe the “normal” movement and differentiate it from a PD patient.
1. Description of experimental research and equipment used, measurement techniques should be more accurate.
R/Thank you for the recommendation. We tried to be precise in our initial description, but we have rewritten this section, using change tracking.
2. Currently, MATLAB was used for the analysis. Are you planning to use other analysis tools?
R/Currently, we are migrating the algorithms to python, using the Anaconda + Jupyter Notebook work environment, because the DSP libraries are open source and can be used by end users without paying Matlab licensing. Additionally, we are working on machine learning techniques. However, for the results shown in the paper, we used Matlab, because was easier and faster to process the results, using the available toolboxes.
Detailed comments:
1. Sections Measurement System and Experimental Setup should more accurately describe the measurement method and the components of the proposed system.
R/Thank you for the recommendation. We tried to be precise in our initial description, but we have rewritten this section, using change tracking.
2. In my opinion, it should be in the caption in Figure 2. Describe exactly what is on each of the four component drawings. Enter points a) ...., b)…,
R/Thank you for the comment. We change the figure and improve the quality, as well as the description.
3. Similarly, for Figures 3,5, and 6 it is necessary to explain what is on the individual components of the figures.
R/Thank you again. We add additional explanation for each one of the figures, trying to improve the legibility of the paper.
4. Please check the literature, it is formatting, and whether I have newer literature item
R/Thank you for the recommendation. We already included 7 new references of articles from 2019 and 2020, addressing the use of wearables for gait analysis.
Reviewer 2 Report
In this work, the authors develop and implement a prototype, obtain interesting results that are compared with those obtained in other methods.
From my point of view this paper is well organized, however I consider that it can still be improved and the following points should be reviewed:
- In the explanation of Figure 2, explain the meaning of "EP" (row 180)
- Figure 2 has very small titles, legends and axis numbering and is not very readable. The same applies to figures 3, 4, 5 and 6
- Refer to where equations (3), (4) and (5) come from. The wording of the text on their origin is not very clear.
- Check "typos" in equations 3 and 4. A comma sign appears.
- Figure 6 improve readability, lines and markers to highlight it.
- Improve wording (rows 242-244).
- Detail and improve the explanation of table 1. Explain in detail (Dx?, Dy?, etc)
Author Response
From my point of view this paper is well organized, however I consider that it can still be improved and the following points should be reviewed:
- In the explanation of Figure 2, explain the meaning of "EP" (row 180)
R/Thank you; was a language mistake. The right is PD (Parkinson’s Disease). It was corrected.
- Figure 2 has very small titles, legends and axis numbering and is not very readable. The same applies to figures 3, 4, 5 and 6
R/Thank you for your comment. We corrected the figure, improving size and legibility of the text.
- Refer to where equations (3), (4) and (5) come from. The wording of the text on their origin is not very clear.
R/Thank you for your comment. We already improve the writing of this section, in order to better specify the origin and our adaptation to our clinical context.
- Check "typos" in equations 3 and 4. A comma sign appears.
R/Thank you very much for your comment. The typos were corrected.
- Figure 6 improve readability, lines and markers to highlight it.
R/Figure 6 was changed to improve readability and a better correlation with equations.
- Improve wording (rows 242-244).
R/Thank you for the comment. We rewrote the paragraph, intending to improve readability.
- Detail and improve the explanation of table 1. Explain in detail (Dx?, Dy?, etc)
R/Thank you. We add some additional lines before table 1, to better explain the table and some terms.